# Technosols Derived from Mining, Urban, and Agro-Industrial Waste for the Remediation of Metal(loid)-Polluted Soils: A Microcosm Assay

**DOI:** 10.3390/toxics11100854

**Published:** 2023-10-12

**Authors:** Antonio Aguilar-Garrido, Ana Romero-Freire, Mario Paniagua-López, Francisco Javier Martínez-Garzón, Francisco José Martín-Peinado, Manuel Sierra-Aragón

**Affiliations:** Departamento de Edafología y Química Agrícola, Facultad de Ciencias, Universidad de Granada, Av. de Fuente Nueva s/n, 18071 Granada, Spain; mpaniagua@ugr.es (M.P.-L.); fjgarzon@ugr.es (F.J.M.-G.); fjmartin@ugr.es (F.J.M.-P.); msierra@ugr.es (M.S.-A.)

**Keywords:** soil pollution, vegetation recovery, potentially toxic elements, waste-derived Technosols, waste valorisation, soil enzymes, toxicity bioassays, *Trifolium campestre*, *Lactuca sativa*

## Abstract

This study evaluated the effectiveness of six Technosols designed for the remediation of polluted soils (PS) by metal(loid)s at physicochemical, biological, and ecotoxicological levels and at a microcosm scale. Technosols T1–T6 were prepared by combining PS with a mix of organic and inorganic wastes from mining, urban, and agro-industrial activities. After two months of surface application of Technosols on polluted soils, we analysed the soil properties, metal(loid) concentration in total, soluble and bioavailable fractions, soil enzymatic activities, and the growth responses of *Trifolium campestre* and *Lactuca sativa in* both the Technosols and the underlying polluted soils. All Technosols improved the unfavourable conditions of polluted soils by neutralising acidity, increasing the OC, reducing the mobility of most metal(loid)s, and stimulating both the soil enzymatic activities and growths of *T. campestre* and *L. sativa*. The origin of organic waste used in the Technosols strongly conditioned the changes induced in the polluted soils; in this sense, the Technosols composed of pruning and gardening vermicompost (T3 and T6) showed greater reductions in toxicity and plant growth than the other Technosols composed with different organic wastes. Thus, these Technosols constitute a potential solution for the remediation of persistent polluted soils that should be applied in large-scale and long-term interventions to reinforce their feasibility as a cost-effective ecotechnology.

## 1. Introduction

Anthropogenic activities, such as metal mining, represent an important potential source of soil pollution [1]. Particularly, the mismanagement of mining wastes (e.g., tailings and acidic waters) can release heavy metals and other associated elements such as arsenic (hereafter referred to as metal(loid)s) into the environment, which can mobilise within the soil–plant–water system, leading to potential adverse effects on the environment and human health [2]. In fact, the number of operating mines has increased considerably in recent decades due to the need for high-priced strategic elements (e.g., rare earth elements, platinum-group elements, and technology-critical elements) [3,4,5]. Thus, the assessment of pollution risk associated with extractive activities and the promotion of prevention, protection, and remediation measures become essential to guarantee the safety of our environment [6].

The effects of pollution accidents, such as the notable example of the Aznalcóllar mine spill (Andalusia, Spain) in 1998, can be long-lasting. For example, the consequences of this tragic environmental accident persist more than 25 years later despite the massive decontamination efforts undertaken by the Regional Government of Andalusia [7], posing a long-term toxic risk to living organisms in the area due to the continued bioaccumulation of metal(loid)s [8]. This potential pollution is revealed by the presence of unvegetated soil patches that are highly heterogeneous in size (with areas between 2 and 200 m^2^) and randomly distributed in the affected area [9]. These patches are characterised by acidic pH levels (around 3.5), high concentrations of metal(loid)s (mainly As and Pb), and low organic carbon contents [10] Appendix A).

The remediation of areas affected by persistent pollution requires specific interventions to reduce their potential toxic risk. In this sense, emerging ecotechnologies such as the use of Technosols specifically designed for a particular environmental problem (called Technosols “a la carte” or “tailor-made” Technosols) [11] can provide viable solutions. This technology is based on pedo-engineering, sustainability, and circular economy approaches, and consists of soils whose properties and pedogenesis are characterised by their technical origin (containing ≥ 20% artifacts by volume) [12]. Indeed, the vast majority of the Technosols applied in soil remediation programmes are waste based, as this ecotechnology also aims to tackle unsustainable waste generation. Besides the general soil functions, Technosols have the capacity to create/stimulate several biogeochemical and edaphic processes (e.g., neutralise acid, decrease sulfide oxidation, immobilise metal(loid)s, increase soil fertility, and stimulate soil biological activity), leading to the sustainability of the rehabilitation process in the medium- and long-term [11,13,14]. The effectiveness of this ecotechnology for the rehabilitation of mining areas (soils, tailings, and leachates) was demonstrated in previous studies for a wide range of Technosols and climate regions both in microcosm studies under controlled conditions [14,15,16,17,18] and large-scale interventions [19,20,21,22,23,24]. However, there are few studies that analyse the changes produced in the physical, chemical, and biological properties at the same time, and this holistic approach is essential to understand the evolution of remediation techniques applied to polluted soils. Likewise, most studies on Technosols focus on analysing the changes in the Technosols themselves and the surface effects, but do not include information on the underlying polluted soils. To the best of our knowledge, this is the first study considering this holistic approach and involving changes produced both in the Technosols and in the underlying polluted soils. The main objective of this study is to evaluate, at physicochemical, biological, and ecotoxicological levels and at a microcosm scale (in growth chamber conditions), the effectiveness of six Technosols produced from polluted soils and a mixture of organic and inorganic wastes in the remediation of metal(loid)-polluted soils; the effects on the underlying polluted soils are also studied as a holistic approach to understand the evolution of this remediation ecotechnology applied to persistent polluted areas.

## 2. Materials and Methods

### 2.1. Polluted Soil: Site Characterisation and Sampling

The polluted soils (PS) by metal(loid)s were sampled in the Guadiamar Green Corridor (southwest Spain), an area affected by one of the largest mining spills in Europe, known as the Aznalcóllar disaster [25]. This area is located in the province of Seville (37°00′–37°30′ N, 6°10′–6°20′ W) and covers a surface that is 45 km long and 0.5 km wide with an irregular shape on both sides of the fluvial plains of the Agrio and Guadiamar rivers. The climate is typically Mediterranean, which is characterised by hot, dry summers and cold, wet winters. The average annual temperature is 18.3 °C, and the annual rainfall is 490 mm, with an evapotranspiration coefficient of 920 mm, based on the past 20 years (2001–2021). The main soil groups in the area are Fluvisols and Regosols, according to the World Reference Base [12], or Entisols, based on Soil Taxonomy—USDA [26]. These are soils with poorly developed profiles, relatively high fertility, and anthropic influences, and are dominated by crops (fruit trees and cereals) and pastures [27].

The soil used was taken from the above-mentioned bare areas that were identified using satellite imagery following the methodology described in [28] (Appendix A). A composite sample was taken at a 0–10 cm depth from 5 different locations to obtain a representative sample of these residually polluted soils (PS). Moreover, a composite sample of a nearby soil unaffected by the spill (hereafter called unpolluted soil (US)) was also taken at a 0–10 cm depth from 5 different sites to serve as a reference.

### 2.2. Production of Technosols

Six Technosols (T1, T2, T3, T4, T5, and T6) were designed and produced by mixing the polluted soil (PS) with a mixture of amendments in the proportions given in Table 1. These amendments consisted of organic and inorganic wastes from mining (iron oxyhydroxide-rich sludge (IO), carbonated waste from peat extraction (CW), and marble cutting and polishing sludge (MS)), urban activity (composted sewage sludge (WS), vermicompost from pruning and gardening (VC)), and agro-industry (solid olive mill by-product (OL)). The selection of these wastes is based on the needs of Technosols to cope with pollution; carbonated wastes (CW and MS) provide pH buffering capacity, organic wastes (WS, VC, and OL) support the recovery of biological activity, and iron-rich wastes (IO) promote the retention capacity of anionic metal(loid)s (e.g., As and Sb). The proportions of wastes in the different Technosols are based on the real doses applied in the field during the remediation actions [29]. The components of the Technosols were dried, sieved to <4 mm, and mixed manually in plastic trays. Technosols were incubated for 2 months at room temperature (20–25 °C) with periodic watering and aeration (every 3 days), maintaining 70% of field capacity. The main physicochemical and biological characteristics of the amendments, as well as their metal(loid) retention capacity in simulated acid mine drainage, as analysed in [30], can be found in Appendix A, respectively.

### 2.3. Experimental Set-Up

The microcosm assay included two controls (polluted soil (PS) and unpolluted soil (US)) and six treatments (T1R1–T6R6) (Appendix A). These treatments consisted of the application of the Technosols (T1–T6) on top of the polluted soil (R1–R6) (Appendix A). Each substrate had six replicates, and each of these replicates consisted of three alveoli of approximately 30 cm^3^ in volume (analytical replicates). All substrates were incubated at 70% of field capacity and at room temperature for one week. After that, each replicate of all substrates was sown with 15 seeds of *Trifolium campestre* Schreb. (5 per alveolus).

### 2.4. Experimental Monitoring and Sample Analysis

The development of *T. campestre* (big hop clover, dicotyledonous, Fabaceae) plants was monitored for two months under controlled conditions and regular watering in growth chambers. Seed germination and mortality of *T. campestre* were examined weekly. At the end of the assay, survival and soil plant analysis development (SPAD) index were measured using a SPAD-502 Plus chlorophyll meter (Konica Minolta Holdings, Inc., Tokyo, Japan). Three leaves from each plant were selected to measure SPAD. Each SPAD value obtained was the mean of 4 readings (2 on each side of the leaf midrib). The SPAD-502 m was calibrated using the reading checker supplied by the manufacturer. The dry biomass of the whole plant was also determined. For this purpose, after harvesting, plant samples were washed with tap water and then with distilled water. After washing, plants were sonicated in distilled water for 30 min. Finally, they were weighed after drying in an oven at 60 °C for 96 h.

After two months of *T. campestre* growth, samples were collected from the polluted soil (PS), unpolluted soil (US), six Technosols (T1–T6), and the underlying polluted soils (R1–R6). In these samples, the following variables were measured: pH in water (1:2.5 m:V); electrical conductivity (EC) in water (1:5 m:V); organic carbon (OC) content via wet oxidation [31]; calcium carbonate (CaCO_3_) content via volumetric gases [32]; total nitrogen (N_T_) and carbon (C_T_) contents via an elemental analyser (LECO^®^ TruSpec CN) (St. Joseph, MI, USA); and metal(loid) concentrations in the total (T), water-soluble (W), and EDTA-extracted bioavailable (E) fractions. These fractions were measured according to the following methodologies: Total concentrations of the studied metal(loid)s (As, Cd, Cu, Pb, Sb, and Zn) were analysed in finely ground samples via X-ray fluorescence (XRF) with a portable NITON XL3t-980 GOLDD+ analyser (Thermo Fisher Scientific, Waltham, MA, Billerica, USA). The precision and accuracy of this method were assessed via the analysis of the certified reference material (CRM 052–050) (RT-Corporation Limited, Salisbury, UK). This analysis gave satisfactory results for all studied elements (Appendix A). The concentration of these metal(loid)s in the water-soluble fraction was determined in a 1:5 (soil/water) suspension according to [33], and the concentration in the bioavailable fraction was extracted using 0.05 M EDTA (pH 7) as described in [34]. The elements extracted from both fractions were measured via inductively coupled plasma mass spectrometry (ICP-MS) in a PerkinElmer^®^ NexION™ 300D spectrometer (Waltham, MA, USA). For quality control of ICP-MS precision and accuracy, a certified reference material (CRM 052–050, RT-Corporation Limited, Salisbury, UK) and procedural blanks were used. In all cases, the measured values were within the confidence interval of the certified value (Appendix A).

In addition, for the ecotoxicological risk assessment, a liquid-phase toxicity bioassay was performed using *Lactuca sativa* L. (lettuce, dicotyledonous, Asteraceae) [35]. In Petri dishes, a filter paper with 5 mL of soil/water extract (1:5) was incubated with 20 seeds per replicate (*n* = 6) at 25 ± 1 °C for 5 days. As a control, the same procedure was conducted with distilled water in triplicate. After this period, the number of germinated seeds and the lengths of developing roots were recorded. The percentage of elongation compared to control samples was calculated, with values from 0% (maximum toxicity) to 100% or higher (no toxicity). In the solid phase, several soil enzymatic activities were also analysed as biological parameters to evaluate the rehabilitation process, namely dehydrogenase [36], β-glucosidase (EC 3.2.1.21) [37], cellulase [38], and acid phosphatase (EC 3.1.3.2) [39]. Dehydrogenase was used as an index of overall microbial activity, while the other enzymatic activities were related to C and P cycles.

### 2.5. Statistical Analysis

Previous to the statistical treatment of the data, the normal distribution test (Kolmogorov–Smirnov) and the homogeneity of variances test (Levene) were performed. As normality and/or homogeneity of variances were not met, non-parametric tests (Kruskal–Wallis and Mann–Whitney U) (*p* < 0.05) were used. To determine the relationship between the different study variables, a principal component analysis (PCA) was performed using Varimax rotation with Kaiser normalisation including soil properties and total, water-soluble, and EDTA-bioavailable concentrations of metal(loid)s, soil enzymatic activities, and endpoints calculated from *T. campestre* and *L. sativa* bioassays. All statistical analyses were carried out using SPSS v. 23.0 software (SPSS Inc., Chicago, IL, USA).

## 3. Results

### 3.1. Soil Properties and Metal(loid) Concentrations

The metal(loid)-polluted soil (PS) had an extremely acidic pH, a very high EC, and a very low CaCO_3_ content. This soil also showed a low potential fertility, manifested by a low OC content, which was five times lower than in the reference unpolluted soil (US), as well as a low total N content (Table 2). Furthermore, it showed high total concentrations of some metal(loid)s (mg kg^−1^; As: 346; Pb: 640) compared to the low concentrations found in the US (mg kg^−1^; As: 25; Pb: 79) (Table 3). Nevertheless, considering the solubility and bioavailability of metal(loid)s, As and Pb were not of critical concern, as the water-soluble (W) and EDTA-bioavailable (E) concentrations were similar to those in the US (Pb_E_ was even lower). In contrast, Cd, Cu, and Zn occurred in very high concentrations in the water-soluble fraction (µg kg^−1^; Cd_W_: 214; Cu_W_: 941; Zn_W_: 11,992), namely 227, 64, and 570 times higher than in the US (Table 3). Additionally, none of the considered metal(loid)s were found in high concentrations in the EDTA-extracted bioavailable fraction. The concentrations were similar to those found in the US, except for Cu and Zn, which were slightly higher.

The characteristics of the Technosols (T1–T6) were significantly better than those of the PS, although there were certain differences between the Technosols (Table 2). All Technosols had a pH between slightly (T2, T5, and T6) and moderately alkaline (T1, T3, and T4) due to the moderately high CaCO_3_ content present (Appendix A). The very high EC in the PS decreased slightly in the Technosols but did not reach the very low values in the US. Thus, the EC remained considerably high in the Technosols. The organic carbon content increased relative to the PS by almost 14-fold in T1, 16-fold in T2, 6.5-fold in T3, 13.5-fold in T4, 14.5-fold in T5, and 5-fold in T6. Thus, the Technosols composed of sewage sludge (T2 and T5) had the highest OC contents, and those composed of vermicompost from pruning and gardening (T3 and T6) had the lowest. The total N in the Technosols was also found in higher concentrations; in T1 and T4 (composed of solid olive mill by-product) and in T2 and T5 (composed of sewage sludge), the concentration of N_T_ was significantly higher than in the US, while in T3 and T6 (composed of vermicompost from pruning and gardening), there were no significant differences compared to the US.

The solubility and bioavailability of metal(loid)s (Table 3) in the Technosols strongly changed in relation to the PS. The very high water-soluble concentrations of Cd, Cu, and Zn in the PS were strongly decreased in all Technosols up to values roughly close to those in the US. In particular, Cd_W_ was diminished by more than 99%, Cu_W_ was diminished by 92 to 98%, and Zn_W_ was diminished by more than 99% in relation to the PS. In contrast, As, Pb, and Sb were not very mobile and bioavailable in the PS, and in the Technosols, their soluble and bioavailable concentrations increased significantly, varying among Technosols. The soluble and bioavailable As contents increased by 3–8 times and 1.4–5.4 times compared to the PS, respectively. The soluble Pb, from being in the PS below the detection limit of ICP-MS, increased to concentrations between 3 and 15 µg kg^−1^ in T1, T2, T4, and T5; however, it remained at undetectable concentrations in T3 and T6. The bioavailable fraction of Pb was similar to the soluble fraction, increasing significantly in T1, T2, T4, and T5 with respect to the PS, while T3 and T6 showed values more similar to the PS. Particularly notable was the increase in the solubility and bioavailability of Sb, as Sb_W_ increased from 7.5-fold to 14.5-fold, and Sb_E_ increased from 4.5-fold to 10-fold in the Technosols with respect to the concentrations in the PS.

The unfavourable initial conditions of the polluted soil (R1–R6) were improved by the surface application of Technosols (Table 4). The pH increased in all underlying polluted soils, but none of them reached a neutral pH. Technosols T1, T2, T4, and T6 were the most effective in buffering the extreme acidity of the PS to slightly acid values in the underlying polluted soils R1, R2, R4, and R6. As for electrical conductivity, it remained at similar values to those of the PS. The organic carbon content also improved due to the effect of Technosols, except in R5 and R6, which remained similar to the PS. In R1, R2, and R4, it almost doubled, and in R3, it also increased but to a lesser extent. Similarly, CaCO_3_ also increased with respect to the PS, especially when treated with marble sludge-derived Technosols (R4, R5, and R6); in the Technosols composed of carbonated waste, it also increased (except R1, where it remained constant), but was not statistically significant. The total C content increased as a direct relation to the rises in OC and CaCO_3_, while the total N remained unchanged.

The effect of the Technosol applications was not reflected in the total metal(loid) concentrations in the underlying polluted soils (R1–R6) but was visible in their solubility/bioavailability (Table 5). Especially noteworthy were the large reductions in the solubility of Cd, Cu, and Zn, as well as the significant increase in the solubility of Sb. However, the effects on bioavailability were not significant for most of the metal(loid)s. The soluble As concentration decreased in all underlying soils compared to the PS, reaching in R2, R3, R4, and R5 values very similar to the US. The bioavailable As showed no significant differences between the treatments and PS except for R2 and R5, where it almost doubled its concentration. The Technosol treatments of the polluted soils were very effective for Cd, Cu, and Zn, reducing their solubility. There was also a significant reduction in the bioavailable fraction of these three elements in the treatments, being more effective in the case of Zn, where the reduction exceeded 50% in all cases. The opposite was observed for Sb in the underlying polluted soils, where both the soluble and bioavailable fractions increased significantly with respect to the PS. However, the increase in the soluble Sb was more pronounced than for the bioavailable fraction, exceeding 3–7 times the soluble concentration in the PS, while for the bioavailable fraction, the increase was not more than twice of that in the PS. The soluble Pb concentrations were below the detection limit of ICP-MS in all cases, and in the bioavailable fraction in the PS, it was very low (0.01 mg kg^−1^), although with the Technosols treatment, it increased slightly but without statistical significance.

### 3.2. Germination and Growth of Trifolium campestre

The treatment of metal(loid)-polluted soil with Technosols (T1R1–T6R6) stimulated the growth of *T. campestre*, compared to no germination of this plant when sown in the polluted soil (PS) (Figure 1). However, not all Technosols showed the same response. The germination rate in the treatments with vermicompost-based Technosols (T3R3 and T6R6) was 50%, similar to that given in the US (57%), while in the others, it was much lower (T1R1: 18%; T2R2: 19%; T4R4: 17%; T5R5: 21%). The survival rate, expressed as the number of *T. campestre* plants established after 10 weeks, showed a similar trend to the germination rate, but with lower values, as some of the germinated seeds died during this time (Figure 1 and Figure 2).

The dry biomass of *T. campestre* in the soil treatments showed differences with respect to the US, which had the highest mean weight (Figure 2). Among the treatments, the highest biomass production was in T3R3 and T6R6, reaching about half of that in the US. In the remaining treatments (T1R1, T2R2, T4R4, and T5R5), the biomass production was significantly lower with no statistical differences among them. The soil plant analysis development (SPAD) index of the *T. campestre* plants was higher in all treatments than in the US, although with a slight variability among them (Figure 2). Indeed, it was only in R1 and R3 that it was statistically significantly superior to the US.

### 3.3. Toxicity Bioassay with Lactuca sativa

The germination rate of *L. sativa* seeds from PS was significantly lower (<50%) than that of the US (>90%). Likewise, elongation in the PS was also much lower than in the US, by about 7.5 times. Both the low germination and elongation of *L. sativa* in the PS were positively stimulated in the Technosols (T1–T6). The germination rate in T2 and T5 remained at values close to those given in the PS, while in the other Technosols, it improved to rates similar to those in the US. However, these differences among the Technosols were not observed for elongation, which was enhanced in all, reaching, in most of them, the elongation given in the US. Indeed, although not statistically significant, the elongation rates in T1 and T4 were slightly higher than that in the US (Figure 3).

Likewise, in the underlying polluted soils (R1–R6), both the germination and elongation rates of *L. sativa* were also improved (Figure 4). The germination rates ranged from 63% to 92%, with the lowest germination rates in R5, and the highest in R2. The increase in elongation was even more considerable, with an elongation in all of them similar to that given in the distilled water control (100%). The maximum elongation was found in R1, which was even higher than in the US, and the minimum elongation was found in R3, although this minimum was at least six times higher than that in the PS.

### 3.4. Soil Enzymatic Activities

The metal(loid)-polluted soil showed low microbiological activity as assessed via enzymatic activities (Figure 5). Dehydrogenase activity, which is widely used as an index of the overall microbial activity, was very low in the PS, while in the US, it was 23 times higher. C-cycle-related enzymes (β-glucosidase and cellulase) were negligible in the PS, as opposed to having almost 100-fold higher values for β-glucosidase and 15-fold higher values for cellulase in the US. The acid phosphatase activity was also reduced by half in the PS compared to the US.

The Technosols showed better soil properties than the PS from which they were produced due to the addition of different wastes, mainly the organic ones (WS, VC, and OL), leading to a stimulation of soil microbiological activity. The increase in dehydrogenase activity was pronounced, especially in the Technosols composed of sewage sludge (T2 and T5) and those composed of solid olive mill by-product (T1 and T4), reaching values above 100 µg TPF g^−1^ in 16 h^−1^, while in the vermicompost-derived Technosols (T3 and T6), the values were about 27 and 11 µg TPF g^−1^ in 16 h^−1^, respectively. As for the rest of the enzymatic activities, improvements were also observed in the Technosols with respect to the PS, but with a large variability among them. In contrast to the dehydrogenase activity, the highest β-glucosidase value was found in the Technosols composed of vermicompost (T3 and T6, 0.8–0.9 p-nitrophenol g^−1^ h^−1^), while in the rest, it was less than half. As for cellulase, the variability was so high that there were no significant differences among the Technosols. However, T4 had the lowest activity, while T2 had the highest (0.23 vs. 0.69 µmol glucose g^−1^ in 16 h^−1^, respectively). The acid phosphatase activity was more than double in the US compared to the PS, and even in T1, T4, and T6, it exceeded the activity measured in the US.

In contrast to the Technosols, the microbiological activity was not recovered in the underlying polluted soils, showing a marked decrease in dehydrogenase activity compared to the Technosols (Figure 6). Little or no significant changes occurred in any of the studied enzymes (dehydrogenase, β-glucosidase, acid phosphatase, and cellulase) in the treated soils (R1–R6) compared to the baseline conditions of the PS. The same pattern of variability in both the dehydrogenase and β-glucosidase activities was repeated in the Technosols (T1–T6) as in the underlying polluted soil (R1–R6). The lowest values of dehydrogenase activity and the highest values of β-glucosidase activity were measured in R3 and R6, although these differences were not statically significant in neither case.

## 4. Discussion

The total soil concentrations of As and Pb in the polluted soil exceed the regulatory levels of 36 mg kg^−1^ and 275 mg kg^−1^ by 9.6 and 2.3 times [40], respectively. However, these total concentrations do not raise much concern, since the risk of leaching of these two elements is very low due to their low solubility [41] and the low precipitation in the area (mean annual precipitation of 490 mm). On the contrary, Cd, Cu, and Zn, with total concentrations not exceeding the regulatory values, showed concentrations high enough to potentially cause toxicity [42,43,44], especially in the water-soluble fraction, which is the most readily accessible to living soil organisms. The bioavailable fraction corresponds to the metal(loid)s contained in the non-silicate-bound soil phases (carbonates and Fe-/Al- oxides), reflecting their availability both in the short-term and relatively long-term [45]. Thus, both the soluble and bioavailable fractions constitute the part of metal(loid)s in soil that can be taken up by organisms and, thus, cause damage to the ecosystem and/or enter the trophic chain [46].

A principal component analysis was conducted to determine the ecotoxicological implications of the changes caused in the metal(loid)-polluted soil after treatment with Technosols. This analysis generated three components that grouped all considered variables (soil properties, metal(loid) fractions, soil enzymatic activities, and bioassay endpoints of *T. campestre* and *L. sativa*), explaining 81.44% of the variance (Appendix A). Component 1 explained 39.76% of the variance and included variables related to potential toxicity (mainly the pH, CaCO_3_ content, and water-soluble and EDTA-bioavailable concentrations of most metal(loid)s) along with the root elongation of *L. sativa* (RE). A direct (positive) relationship between the pH and RE, as well as an inverse (negative) relationship between the pH and solubility and bioavailability of most metal(loid)s (with the exception of As and Sb) were observed. Component 2, explaining 25.85% of the variance, grouped the EC with the total concentrations of most metal(loid)s and the water-soluble concentrations of Cd, Cu, and Zn in a direct relationship; in contrast, the EC was inversely related to the C-cycle enzymes (β-glucosidase and cellulase), biomass and survival of *T. campestre*, and root elongation of *L. sativa*. Component 3 explained 15.83% of the variance and showed that the soil OC is directly related to the total N and C, and also to dehydrogenase activity.

In this sense, although the total concentrations of metal(loid)s decreased significantly in Technosols due to the dilution effect, they still exceeded the regulatory levels for As and Pb. However, the solubility and bioavailability of the metal(loid)s strongly changed in both Technosols (T1–T6) and in the underlying polluted soils (R1–R6). The high acidity of polluted soil, caused by the formation of sulphuric acid in the soil matrix from the oxidation of remaining pyritic sludge [47], was neutralised in the Technosols via the reaction with carbonates provided by the carbonate-rich wastes (MS and CW) [48], while in the underlying soils, the leachates coming from the Technosols increased the pH by 2 to 3 units in relation to the PS, raising the values to near neutrality. The oxidation of pyritic sludge also leads to the formation of soluble sulphates that increase the EC in polluted soil [49]. The increases in acidity and salinity due to the oxidation of sulphides are correlated with the higher solubility of more mobile metals such as Cd, Cu, and Zn [50,51,52,53] with respect to the less soluble ones (As, Pb, and Sb) (Appendix A). In this sense, the most mobile elements are significantly reduced in both Technosols and underlying soils with respect to the PS, but in the underlying soil, these soluble concentrations are higher due to potential leaching effects, coinciding with previous observations under natural conditions [54].

On the contrary, these less soluble metal(loid)s (As, Pb, and Sb) can be desorbed by the competition between arsenates (AsO_4_^3−^) and organic matter for the adsorption positions of iron oxides [55,56]; thus, the As resolubilisation that occurred in the Technosols with a significantly higher OC content was not produced in the underlying polluted soils. Additionally, the influence of pH on As mobility is a key factor. In our case, a lower adsorption of As was observed when the pH rose, and this was attributable to the more negatively charged arsenate species repulsing the anion exchange sites [47]. Otherwise, under the irrigation conditions during our experiment, a reductive dissolution of iron oxides (e.g., ferrihydrite) may result in the substantial mobilisation of As in a pH range between 6 and 8 [57]. Like As, Sb solubility is also strongly controlled by the pH and soil organic matter (SOM) content (Appendix A), so Sb solubilisation/desorption with a decreasing acidity as well as increasing soil organic matter content may be related to competition in the formation of Fe-SOM-Sb complexes [58].

Likewise, both in the Technosols (T1–T6) and in the underlying polluted soils (R1–R6), decreases in the bioavailable fractions of Cd, Cu, and Zn were observed in relation to the PS, possibly due to the increase in pH and the presence of calcium carbonate from the carbonated wastes (CW and MS). These elements in acidic soils are usually adsorbed in non-specific forms and, therefore, in reversible processes, while in neutral or alkaline soil, they can be specifically adsorbed or even occluded by iron oxides and hydroxides [59,60], or co-precipitated as an iron-sulphate and hydroxysulfate complex [61], which are considered as less reversible reactions. However, the bioavailable fractions of As and Sb were not reduced with respect to the initial polluted soil, neither in the Technosols nor in the underlying soils. This is possibly related to the formation of organic–metallic complexes, which could lead to an increase in the availability of these elements when the pH rises [58,62].

The microcosm assay with *Trifolium campestre* Schreb. evaluated the toxic effect directly in the soil fraction [63]. This plant was selected for its capacity to cope with soil pollution and its spontaneous presence in the study area of the Guadiamar Green Corridor [41]. Based on this assay, it is inferred that the use of Technosols on polluted soils (T1R1–T6R6) can also be considered to decrease the ecotoxicological risk in the area, as it allows for an optimal development of *T. campestre*. The growth stimulation of *T. campestre* was significant in all treatments, with a high survival rate, biomass, and SPAD index compared to the polluted soil. These results are consistent with those reported for other plant species (e.g., *Eucalyptus globulus* Labill., *Cistus ladanifer*, *Dactylis glomerata* L., *Erica australis* L., and *Lablab purpureus* (L.) Sweet.) grown in different Technosols also composed of mining waste/polluted soils, either in field or greenhouse assays [64,65,66,67]. Plant growth can be related to the improvement of some soil properties (pH neutralisation and increases in OC, CaCO_3_, and N_T_), microbiological activity, and decreases in the solubility and bioavailability of most metal(loid)s both in Technosols (T1–T6) and in underlying polluted soil (R1–R6) [20]. However, treatments T3R3 and T6R6, in which Technosols composed of pruning and gardening vermicompost were used, showed the best performance, indicating that the nature of the organic wastes influences the capacity to develop a vegetation cover [68,69,70].

In general, all treatments had a similar response in the toxicity bioassay with *L. sativa*, with SG and RE values similar to those of the unpolluted soil (except RE at T2 and T5). The higher toxicity at T2 and T5, represented by a low germination, may be related to the type of organic waste used; in this case, sewage sludge, despite being widely used as an amendment, usually contains a complex mixture of pollutants (e.g., heavy metals and emerging organic pollutants such as pharmaceuticals) that can be harmful to organisms [71].

The effectiveness of the rehabilitation process was also assessed by determining the soil enzymatic activities, which are widely used as biological indicators that reflect the soil functional diversity, changes in the microbial community composition, and microbial status [72]. In this sense, the Technosols improved the soil quality and largely neutralised metal(loid) pollution, strongly stimulating microbiological activity, indicating the good performance of all microbial communities involved in organic matter degradation, mineralisation processes, and nutrient cycling [24]; however, this improvement in the enzymatic activity did not occur in the underlying polluted soils, indicating that the changes in some physicochemical properties and the time elapsed during the experiment were not enough to promote microbial activity in the treated soils.

The addition of organic and inorganic amendments to degraded environments such as these polluted soils leads to a boost in the soil microorganism’s activity, evaluated via the dehydrogenase activity [65]. In this sense, the increase in the OC content in the Technosols was much greater than in the underlying polluted soils, which is directly related to the dehydrogenase activity (Appendix A). This is in agreement with several studies [14,65,72] that report that an increase in the OC content due to organic amendment addition is a key driver of microbiological activity. However, it is not only the OC content that determines the biological activity, but also the different nature of organic matter [73,74,75]. Thus, while T3 and T6 were the Technosols with the lowest OC contents and dehydrogenase activities, they had the highest β-glucosidase activity of all the tested Technosols, which may be due to the fact that the pruning and gardening vermicompost has a higher proportion of more easily decomposable compounds (e.g., hemicellulose and cellulose) than more resistant ones (e.g., lignin) compared to the other organic wastes used [76].

## 5. Conclusions

Soils with persistent pollution in the Guadiamar Green Corridor (more than 25 years after the Aznalcóllar mine accident) constitute degraded environments characterised by high acidity and salinity, low fertility, and high concentrations of metal(loid)s in total, soluble, and bioavailable fractions. These conditions cause the absence of vegetation and low soil enzymatic activity. Thus, this study can provide valuable information for the study of remediation solutions for metal(loid)-polluted soils, based on the ecotechnology of Technosols, and then serve as a model for consultation.

The six designed Technosols constructed from mixing the polluted soil with a combination of organic and inorganic wastes from local industries (mines, urban gardening services, wastewater treatment services, and olive mills) were effective in soil remediation over the time span tested (2 months) under growth chamber conditions. In general, all Technosols improved the unfavourable conditions of polluted soils (by neutralising the acidity and increasing the OC), reduced the solubility/bioavailability of most metal(loid)s (with the exception of As and Sb, and Pb in some Technosols), and strongly stimulated soil enzymatic activity and the growths of *T. campestre* and *L. sativa* by reducing their potential toxicities. In particular, the Technosols composed of pruning and gardening vermicompost (T3 and T6) showed the best overall response, while those composed of sewage sludge (T2 and T5) showed the worst performance, especially in terms of toxicity reduction, as evaluated using bioassays with *T. campestre* and *L. sativa*. This highlights the importance of the organic matter nature in the organic wastes used in Technosols, as it greatly influences the improvement in the soil conditions, metal(loid)s’ mobility, microbiological activity, and, thus, its capacity to develop a vegetation cover. The Technosols also improved the physicochemical properties and reduced the mobility of most metal(loid)s of the underlying polluted soils, although the biological activity evaluated via enzymatic activity was not significantly modified over the time of the experiment. Thus, it was shown that these Technosols can constitute a potential solution for the remediation of persistent polluted soils; nevertheless, they should be applied and assessed in large-scale and long-term interventions to reinforce their feasibility as a cost-effective ecotechnology.

## Figures and Tables

**Figure 1 toxics-11-00854-f001:**
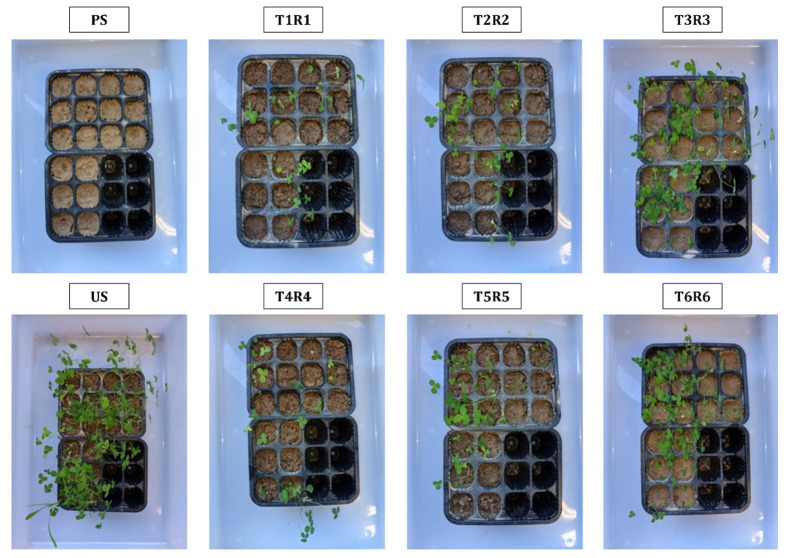
General appearance of *T. campestre* plants after two months of growth with polluted soil (PS), unpolluted soil (US), and the treatments consisting of the application of each Technosol on top of the polluted soil (T1R1–T6R6).

**Figure 2 toxics-11-00854-f002:**
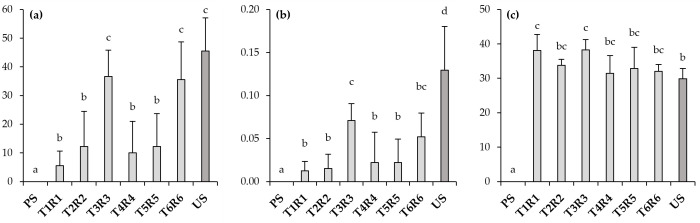
Survival (**a**), dry biomass weight (**b**), and soil plant analysis development (SPAD) index (**c**) of *T. campestre* plants after two months of growth on polluted soil (PS), unpolluted soil (US), and the treatments consisting of the application of each Technosol on top of the polluted soil (T1R1–T6R6). Letters represent significant differences among different materials (Kruskal–Wallis and Mann–Whitney U tests; *p* < 0.05).

**Figure 3 toxics-11-00854-f003:**
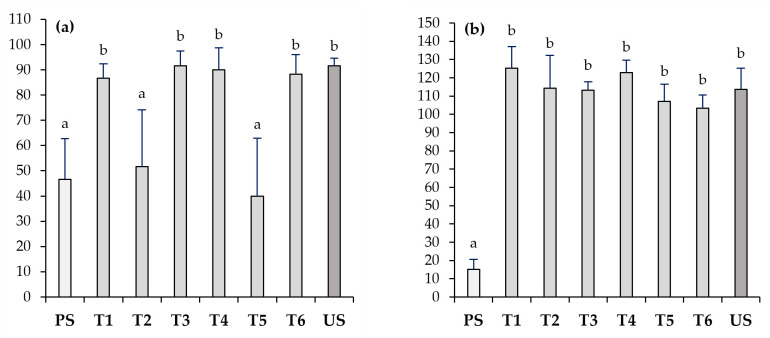
Seed germination (SG) (**a**) and root elongation (RE) (**b**) of *L. sativa* in polluted soil (PS), unpolluted soil (US), and six designed Technosols (T1–T6) (*n* = 6). Letters represent significant differences among different materials (Kruskal–Wallis and Mann–Whitney U tests; *p* < 0.05).

**Figure 4 toxics-11-00854-f004:**
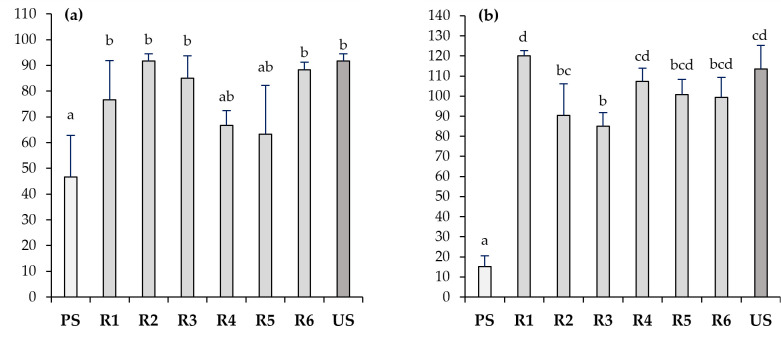
Seed germination (SG) (**a**) and root elongation (RE) (**b**) of *L. sativa* in polluted soil (PS), unpolluted soil (US), and polluted soils treated with application of each Technosol (R1–R6) (*n* = 6). Letters represent significant differences among different materials (Kruskal–Wallis and Mann–Whitney U tests; *p* < 0.05).

**Figure 5 toxics-11-00854-f005:**
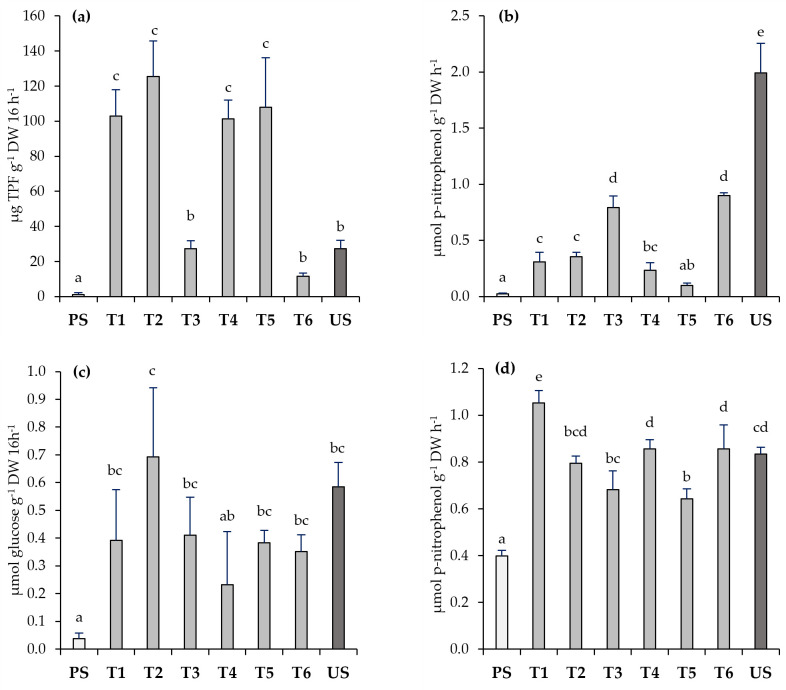
Enzymatic activities (dehydrogenase (**a**), β-glucosidase (**b**), cellulase (**c**), and acid phosphatase (**d**)) in polluted soil (PS), unpolluted soil (US), and six designed Technosols (T1–T6) (*n* = 6). Letters represent significant differences among different materials (Kruskal–Wallis and Mann–Whitney U tests; *p* < 0.05).

**Figure 6 toxics-11-00854-f006:**
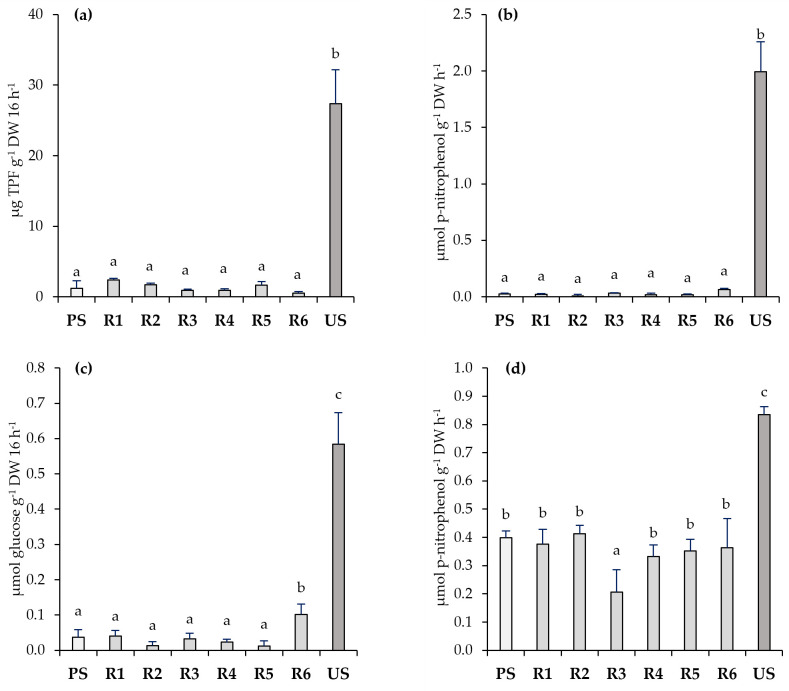
Enzymatic activities of dehydrogenase (**a**), β-glucosidase (**b**), cellulase (**c**), and acid phosphatase (**d**) in polluted soil (PS), unpolluted soil (US), and polluted soils treated with the application of each Technosol (R1-R6) (*n* = 6). Letters represent significant differences among different materials (Kruskal–Wallis and Mann–Whitney U tests; *p* < 0.05).

**Table 1 toxics-11-00854-t001:** Composition of Technosols: % of polluted soil and each waste.

Technosols	PS	IO	CW	MS	OL	WS	VC
T1	60	2	20	-	18	-	-
T2	60	2	20	-	-	18	-
T3	60	2	20	-	-	-	18
T4	60	2	-	20	18	-	-
T5	60	2	-	20	-	18	-
T6	60	2	-	20	-	-	18

Polluted soil (PS), iron oxyhydroxide-rich sludge (IO), carbonated waste from peat extraction (CW), marble cutting and polishing sludge (MS), solid olive mill by-product (OL), composted sewage sludge (WS), vermicompost from pruning and gardening (VC).

**Table 2 toxics-11-00854-t002:** Physicochemical properties of polluted soil (PS), unpolluted soil (US), and six designed Technosols (T1–T6) (mean ± SD; *n* = 6).

	PS	US	Technosols					
			T1	T2	T3	T4	T5	T6
pH (H_2_O) 1:2.5	3.53 ± 0.03 a	6.91 ± 0.08 b	8.19 ± 0.04 e	7.83 ± 0.02 c	8.10 ± 0.03 d	8.01 ± 0.05 d	7.79 ± 0.01 c	7.78 ± 0.04 c
EC 1:5 (dS m^−1^)	2.77 ± 0.12 d	0.05 ± 0.01 a	2.09 ± 0.15 c	1.37 ± 0.16 b	1.25 ± 0.22 b	2.18 ± 0.25 c	1.62 ± 0.25 b	1.42 ± 0.19 b
OC (%)	0.42 ± 0.11 a	2.24 ± 0.14 bc	5.97 ± 0.15 d	6.61 ± 0.31 e	2.74 ± 0.18 c	5.67 ± 0.33 d	6.15 ± 0.37 de	2.15 ± 0.12 b
CaCO_3_ (%)	0.39 ± 0.07 a	2.66 ± 0.05 b	27.22 ± 4.48 c	26.11 ± 1.48 c	26.87 ± 0.97 c	27.57 ± 2.03 c	26.85 ± 4.31 c	25.92 ± 2.59 c
N_T_ (%)	0.11 ± 0.01 a	0.20 ± 0.02 bc	0.52 ± 0.01 d	0.84 ± 0.04 e	0.22 ± 0.02 c	0.51 ± 0.03 d	0.90 ± 0.03 f	0.14 ± 0.03 ab
C_T_ (%)	0.48 ± 0.02 a	2.50 ± 0.19 b	8.28 ± 0.13 d	8.40 ± 0.16 d	5.20 ± 0.20 c	8.66 ± 0.29 d	8.71 ± 0.19 d	5.02 ± 0.07 c

EC—electrical conductivity; OC—organic carbon content; CaCO_3_—calcium carbonate content; N_T_/C_T_—total N and C concentrations. Letters represent significant differences among different materials (Kruskal–Wallis and Mann–Whitney U tests; *p* < 0.05).

**Table 3 toxics-11-00854-t003:** Total, water-soluble, and EDTA-extracted bioavailable concentrations of metal(loid)s in polluted soil (PS), unpolluted soil (US), and six designed Technosols (T1–T6) (mean ± SD; *n* = 6).

	PS	US	Technosols					
			T1	T2	T3	T4	T5	T6
Total (T) (mg kg^−1^)								
As	345.66 ± 25.29 f	25.02 ± 2.71 a	215.73 ± 6.12 b	242.76 ± 4.25 c	300.88 ± 10.03 e	250.33 ± 3.60 cd	205.49 ± 2.87 b	268.79 ± 4.45 d
Cu	106.80 ± 5.24 c	37.72 ± 2.00 a	72.01 ± 2.68 b	109.76 ± 12.08 c	73.93 ± 3.60 b	72.91 ± 4.21 b	101.43 ± 3.91 c	64.09 ± 1.65 b
Pb	640.17 ± 48.69 e	79.17 ± 10.76 a	382.61 ± 8.80 b	431.95 ± 12.92 bc	511.50 ± 6.16 d	428.24 ± 6.45 bc	366.31 ± 2.40 b	474.56 ± 5.40 cd
Sb	20.83 ± 5.10 a	n.d.	21.91 ± 4.18 a	23.10 ± 5.32 a	19.68 ± 5.75 a	19.92 ± 4.72 a	23.61 ± 2.73 a	23.06 ± 6.12 a
Zn	185.09 ± 6.46 d	138.13 ± 1.54 c	106.05 ± 5.86 a	303.55 ± 1.34 f	144.66 ± 2.96 c	111.91 ± 0.33 a	266.93 ± 2.15 e	124.32 ± 3.02 b
Water-soluble (W) (µg kg^−1^)								
As	8.91 ± 1.41 a	4.50 ± 2.86 a	47.51 ± 8.45 c	37.00 ± 2.43 bc	71.01 ± 2.78 d	67.40 ± 15.99 d	39.47 ± 4.34 bc	26.54 ± 0.34 b
Cd	213.89 ± 42.71 b	0.94 ± 0.63 a	0.59 ± 0.05 a	1.02 ± 0.02 a	0.43 ± 0.06 a	1.26 ± 0.53 a	0.69 ± 0.09 a	0.25 ± 0.04 a
Cu	940.79 ± 105.12 d	14.60 ± 4.10 a	37.45 ± 3.70 a	76.56 ± 2.20 a	24.24 ± 5.16 a	75.46 ± 20.51 a	85.80 ± 9.55 a	14.43 ± 0.92 a
Pb	n.d.	n.d.	3.05 ± 2.63 a	8.20 ± 2.77 ab	n.d.	9.60 ± 5.78 b	15.97 ± 1.80 c	n.d.
Sb	27.01 ± 1.65 a	19.69 ± 5.17 a	308.80 ± 4.68 d	224.38 ± 2.82 b	395.43 ± 17.06 e	262.03 ± 20.88 c	204.34 ± 25.10 b	317.70 ± 14.77 d
Zn	11,992.27 ± 2953.45 b	21.04 ± 11.47 a	14.39 ± 3.56 a	36.25 ± 3.32 a	3.83 ± 0.90 a	39.07 ± 20.96 a	51.21 ± 4.86 a	4.54 ± 1.06 a
EDTA-extracted (E) (mg kg^−1^)								
As	0.18 ± 0.03 b	0.11 ± 0.02 a	0.98 ± 0.05 g	0.57 ± 0.01 e	0.91 ± 0.02 f	0.40 ± 0.07 d	0.34 ± 0.01 d	0.26 ± <0.01 c
Cd	0.30 ± 0.09 b	0.32 ± 0.10 b	0.09 ± <0.01 a	0.10 ± <0.01 a	0.10 ± <0.01 a	0.11 ± <0.01 a	0.11 ± <0.01 a	0.11 ± 0.01 a
Cu	5.16 ± 0.37 d	2.09 ± 0.07 c	2.09 ± 0.02 c	1.65 ± 0.01 b	1.38 ± 0.03 a	2.39 ± 0.24 c	2.03 ± 0.08 c	1.74 ± 0.06 b
Pb	0.01 ± <0.01 a	6.56 ± 0.18 c	0.39 ± 0.14 b	0.61 ± <0.01 b	0.09 ± 0.01 a	0.82 ± 0.43 b	0.68 ± 0.02 b	0.09 ± <0.01 a
Sb	0.13 ± 0.02 a	0.22 ± 0.02 b	1.35 ± 0.02 h	0.88 ± 0.01 e	1.17 ± 0.02 g	0.78 ± 0.06 d	0.60 ± 0.02 c	0.95 ± 0.03 f
Zn	12.99 ± 7.05 bc	7.70 ± 3.64 b	2.09 ± 0.02 a	7.11 ± 0.12 b	2.54 ± 0.03 a	1.79 ± 0.02 a	8.80 ± 0.21 b	2.12 ± 0.04 a

Cd_T_: non-detectable (n.d.). Letters represent significant differences among different materials (Kruskal–Wallis and Mann–Whitney U tests; *p* < 0.05).

**Table 4 toxics-11-00854-t004:** Physicochemical properties of polluted soil (PS), unpolluted soil (US), and polluted soils treated with the application of each Technosol (R1–R6) (mean ± SD, *n* = 6).

	PS	US	Polluted Soil Treated with Each Technosol					
			R1	R2	R3	R4	R5	R6
pH (H_2_O) 1:2.5	3.53 ± 0.03 a	6.91 ± 0.08 e	6.51 ± 0.21 d	6.41 ± 0.18 d	5.47 ± 0.47 b	6.09 ± 0.21 cd	5.90 ± 0.35 bc	6.08 ± 0.25 cd
EC 1:5 (dS m^−1^)	2.70 ± 0.27 bcd	0.05 ± 0.01 a	2.72 ± 0.07 cd	2.49 ± 0.08 b	2.54 ± 0.03 bc	2.72 ± 0.05 cd	2.59 ± 0.12 bcd	2.78 ± 0.14 d
OC (%)	0.42 ± 0.11 a	2.24 ± 0.14 c	1.03 ± 0.25 b	0.98 ± 0.12 b	0.77 ± 0.09 b	0.96 ± 0.07 b	0.62 ± 0.26 ab	0.51 ± 0.05 a
CaCO_3_ (%)	0.39 ± 0.07 a	2.66 ± 0.05 c	0.40 ± 0.08 a	0.58 ± 0.26 a	0.66 ± 0.15 a	1.19 ± 0.25 b	1.18 ± 0.22 b	1.26 ± 0.29 b
N_T_ (%)	0.11 ± 0.01 a	0.20 ± 0.02 b	0.10 ± 0.01 a	0.11 ± 0.01 a	0.09 ± 0.01 a	0.11 ± 0.05 a	0.13 ± 0.01 a	0.10 ± 0.01 a
C_T_ (%)	0.48 ± 0.11 a	2.50 ± 0.19 e	1.22 ± 0.12 d	1.18 ± 0.05 cd	0.98 ± 0.04 bc	1.36 ± 0.06 cd	0.90 ± 0.12 d	0.72 ± 0.05 bc

EC—electrical conductivity; OC—organic carbon content; CaCO_3_—calcium carbonate content; N_T_/C_T_—total N and C concentrations. Letters represent significant differences among different materials (Kruskal–Wallis and Mann–Whitney U tests; *p* < 0.05).

**Table 5 toxics-11-00854-t005:** Total, water-soluble, and EDTA-extracted bioavailable concentrations of metal(loid)s in polluted soil (PS), unpolluted soil (US), and polluted soils treated with the application of each Technosol (R1–R6) (mean ± SD; *n* = 6).

	PS	US	Polluted Soils Treated with Each Technosol					
			R1	R2	R3	R4	R5	R6
Total (T) (mg kg^−1^)								
As	345.66 ± 25.29 b	25.02 ± 2.71 a	346.58 ± 13.46 b	359.32 ± 13.24 b	361.52 ± 7.39 b	345.69 ± 38.61 b	356.70 ± 10.15 b	347.20 ± 16.39 b
Cu	106.80 ± 5.24 b	37.72 ± 2.00 a	101.06 ± 3.58 b	105.77 ± 5.10 b	105.48 ± 11.42 b	100.88 ± 13.43 b	101.59 ± 8.69 b	105.93 ± 8.28 b
Pb	640.17 ± 48.69 b	79.17 ± 10.76 a	606.36 ± 15.42 b	615.74 ± 34.93 b	628.73 ± 16.95 b	597.00 ± 57.24 b	612.49 ± 17.27 b	596.33 ± 24.51 b
Sb	20.83 ± 5.10 a	n.d.	24.11 ± 3.81 a	21.30 ± 6.32 a	19.98 ± 7.75 a	19.49 ± 6.57 a	22.16 ± 3.72 a	22.78 ± 5.40 a
Zn	185.09 ± 6.46 b	138.13 ± 1.54 a	172.37 ± 7.15 b	183.44 ± 5.95 b	184.38 ± 6.27 b	186.71 ± 8.09 b	183.12 ± 3.80 b	177.49 ± 4.97 b
Water-soluble (W) (µg kg^−1^)								
As	8.91 ± 1.41 a	4.50 ± 2.86 a	7.39 ± 2.16 a	4.84 ± 0.51 a	4.87 ± 1.56 a	4.42 ± 1.45 a	4.51 ± 0.73 a	5.15 ± 2.68 a
Cd	213.89 ± 42.71 c	0.94 ± 0.63 a	3.07 ± 1.55 a	11.09 ± 1.50 ab	37.58 ± 9.91 b	7.43 ± 2.41 a	6.67 ± 2.81 a	6.35 ± 2.45 a
Cu	940.79 ± 105.12 c	14.60 ± 4.10 a	73.25 ± 8.76 ab	127.01 ± 14.35 b	52.76 ± 10.53 ab	39.65 ± 1.29 ab	114.72 ± 13.68 b	34.36 ± 7.31 ab
Sb	27.01 ± 1.65 a	19.69 ± 5.17 a	188.50 ± 50.49 c	105.24 ± 22.39 b	76.07 ± 23.53 ab	89.48 ± 19.78 b	99.24 ± 24.69 b	93.91 ± 4.95 b
Zn	11,992.3 ± 1652.7 b	21.04 ± 6.96 a	44.60 ± 32.18 a	425.03 ± 229.73 a	1766.10 ± 802.83 a	214.53 ± 116.85 a	186.36 ± 94.10 a	60.55 ± 26.82 a
EDTA-extracted (E) (mg kg^−1^)								
As	0.18 ± 0.03 bc	0.11 ± 0.02 a	0.22 ± 0.02 c	0.37 ± 0.04 d	0.21 ± <0.01 c	0.14 ± 0.01 ab	0.34 ± 0.05 d	0.14 ± 0.01 ab
Cd	0.30 ± 0.09 bc	0.32 ± 0.10 c	0.14 ± 0.04 a	0.16 ± 0.02 ab	0.18 ± 0.01 ab	0.13 ± 0.01 a	0.13 ± 0.02 a	0.20 ± 0.03 abc
Cu	5.16 ± 0.37 c	2.09 ± 0.07 a	3.62 ± 0.20 b	3.70 ± 0.71 b	4.20 ± 0.06 b	4.14 ± 0.12 b	3.31 ± 0.07 b	3.63 ± 0.46 b
Pb	0.01 ± <0.01 a	6.56 ± 0.18 b	0.02 ± <0.01 a	0.08 ± 0.03 a	0.02 ± <0.01 a	0.03 ± 0.01 a	0.13 ± 0.04 a	0.03 ± <0.01 a
Sb	0.13 ± 0.02 a	0.22 ± 0.02 b	0.26 ± 0.03 b	0.24 ± 0.02 b	0.20 ± 0.02 b	0.22 ± 0.01 b	0.23 ± 0.04 b	0.20 ± 0.03 b
Zn	12.99 ± 4.63 b	7.70 ± 3.64 ab	4.16 ± 1.70 a	5.09 ± 0.89 a	5.14 ± 0.45 a	3.05 ± 0.38 a	3.66 ± 0.78 a	5.23 ± 0.85 a

Cd_T_, Pb_W_: non-detectable (n.d.). Letters represent significant differences among different materials (Kruskal–Wallis and Mann–Whitney U tests; *p* < 0.05).

## Data Availability

The data presented in this study are available upon request from the corresponding authors.

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
