# Peer review of "Technosols Derived from Mining, Urban, and Agro-Industrial Waste for the Remediation of Metal(loid)-Polluted Soils: A Microcosm Assay"

_toxics, 2023, doi:10.3390/toxics11100854_

Round 1

Reviewer 1 Report

The remediation of  soils in mining zones and the surrouding is important. The manuscript provide the report about possibility of use 6 Technolsol to control the issue, which contains  valuable data including the physico-chemical-biological examination. It will help to devolpe the control approach for contaminated soils. However, the manuscript needs more illustration of its scientific value.

The author may provide a better abstract with more specific and strong scientic statement of main findings.

Introduction part is lack of indication of related research gap and novelty of this research.

The reason of selection of those materials for technolsol is not clear provided. Also, how do the author set the adding portion of the materials in 6 technolsol?

The conc. of Sb in PS is far lower than other pollutants. Why do the author consider Sb as research target? What is the regulatory values for Sb and other elements?

The water-soluble and EDTA extracted As are higher in Technolsol than PS, while higher EDTA extracted As are found in R serial soil. It was sad to find the results since there are large ratio of input of Fe containing agents in several Technolsol. May the author provide more discussion about it?  Do the increase of SOC show so remarkable competition than As for adsorption sites?

It is good to show the picture of plant growth. But the repeatability of germination and growth are indeed poor. Also the error bar in Figure 2 etc. are very big. Also, the survival of T.campestre in US treatment is lower than 50%. Although the difference between PS and amended soil is remarkable, I am worry about the accuracy and credibility of the plant experiment. 

Section 3.4 maybe better move ahead to 3.2. In my opinion, the toxicity test shall be done before the planation experiment.

The discussion section shall be well refined. Is there necessarity to describe the soil pollution and the importance of test of soluble/bioavailable metal using as long as two paragraph ? Instead of, the illustration of the interaction of components of technolsol with T soil or R soil (both basic properties and metal availability ) is not sufficient. Also, the changs of R soil by the top T soil needs in-depth discussion. In short words, the plausible mechanism of T soil is not well presented, and the comparison of the effectiveness and interaction mechanism among 6 T soils shall be strenghthened. 

There are as high as 106 references. May the author refine it by remove the out-of-date or less relevant literatures ?

The writting is acceptable. Checked the detailed presentation. Like "p < 0.05", the letter p is better italic.

Author Response

Please find attached the response to your comments

Reviewer 2 Report

1. The abstract of this paper is well written, including the research purpose, research methods and research results.

2.The introduction is written with a bit of background, hoping that the author can give a little description based on his own research results, such as what kind of research results can eventually form what kind of practical application. The profile should be closely related to the results.

3. The conclusion is promising, but the connection with the discussion is not smooth enough, the short explanation of the conclusion is lacking, and the writing is a little messy. I hope to briefly summarize some of the original conclusions of the use of technical sol for the remediation of metal contaminated soil, and set off your research value.

4. There are a lot of data and analysis, you need to specify the key information to be discussed and concluded.

5. In the discussion section, please ask someone who is proficient in English to review and polish the original manuscript, which will make the paper more readable.

6. In this paper, the data in Table 5 can be briefly summarized and analyzed, and then some relevant conclusions can be drawn.

7. In general, the manuscript was prepared without due stylistic, grammatical, and punctuation care, and thus with a certain disregard for the reader. Although I am not a native speaker, however, in my opinion, the text of the paper could benefit from a careful revision to improve English grammar and spelling prior to publishing.

-

Author Response

Please find attached the response to your comments.

Reviewer 3 Report

It is an interest study ad well presented

Author Response

Dear Reviewer 3,
Thank you for your valuable comments on our manuscript ID toxics-2628356 entitled “Technosols derived from mining, urban, and agro-industrial waste for the remediation of metal(loid)s-polluted soils: A microcosm assay” to the Special Issue “Soil and Water Pollution, Remediation and Ecotoxicity Assessment” of the journal Toxics. We appreciate the time and effort you have spent in reviewing our paper.

Reviewer 4 Report

Manuscript presented a detailed study on the remediation of metal(loid)-polluted soils using Technosols derived from various waste materials. The study is grounded on a well-established problem and leverages a comprehensive set of analyses to assess the potential of Technosols in soil remediation.

1) How and why were such proportions of components selected in Technosols?

2) Could you discuss the implications of the increased solubility of As and Sb in some Technosols?

3) How do the results of the plant bioassays correlate with the changes in soil properties and metal(loid) concentrations?

Author Response

(The authors gave the same response as above.)

Round 2

Reviewer 1 Report

The revised version can be considered for publication.